# Invasiveness, Monitoring and Control of *Hakea sericea*: A Systematic Review

**DOI:** 10.3390/plants12040751

**Published:** 2023-02-07

**Authors:** Tamiel Khan Baiocchi Jacobson, Dionatan Gerber, João Carlos Azevedo

**Affiliations:** 1Centro de Investigação de Montanha (CIMO), Instituto Politécnico de Bragança, Campus Santa Apolónia, 5300-253 Bragança, Portugal; 2Faculdade UnB Planaltina, UnB/FUP—Universidade de Brasília, Brasília 73345-010, Brazil; 3Centro de Ecologia Funcional (CFE), Departamento de Ciência da Vida, Universidade de Coimbra, 300-456 Coimbra, Portugal; 4Departamento de Ciências Florestais e Arquitetura Paisagista, Universidade de Trás-os-Montes e Alto Douro, 5001-801 Vila Real, Portugal; 5Instituto de Investigação Interdisciplinar, Universidade de Coimbra, 3004-531 Coimbra, Portugal; 6Laboratório Associado para a Sustentabilidade e Tecnologia em Regiões de Montanha (SusTEC), Instituto Politécnico de Bragança, Campus de Santa Apolónia, 5300-253 Bragança, Portugal

**Keywords:** biological invasion, alien species, Mediterranean ecosystems, ecosystem services, control management, fire

## Abstract

Solutions for ecological and economic problems posed by *Hakea sericea* invasions rely on scientific knowledge. We conducted a systematic review to analyze and synthesize the past and current scientific knowledge concerning *H. sericea* invasion processes and mechanisms, as well as monitoring and control techniques. We used ISI Web of Science, Scopus, and CAPES Periodicals to look for publications on the ecological and environmental factors involved in *H. sericea* establishment (question 1); responses of *H. sericea* to fire in native and invaded ecosystems (question 2); and *H. sericea* monitoring and control methods (question 3). We identified 207 publications, 47.4% of which related to question 1, mainly from Australia and South Africa, with an increasing trend in the number of publications on monitoring and modeling. The traits identified in our systematic review, such as adaptations to dystrophic environments, drought resistance, sclerophylly, low transpiration rates, high nutrient use efficiency, stomatal conductance and photosynthetic rates, strong serotiny, proteoid roots and high post-fire seed survival and seedling recruitment, highlighted that *H. sericea* is a successful invader species due to its long adaptive history mediated by an arsenal of ecophysiological mechanisms that place it at a superior competitive level, especially in fire-prone ecosystems. Integrated cost-effective control methods in selected areas and the incorporation of information on the temporal invasion dynamics can significantly improve invasion control and mitigate *H. sericea* impacts while maintaining the supply of ecosystem services in invaded areas.

## 1. Introduction

*Hakea sericea* Schrad. & J.C.Wendl (Proteaceae) (synonyms: *Banksia acicularis*, *Banksia tenuifolia*, *Conchium aciculare*, *Conchium compressum*, *Hakea acicularis*, *Hakea acicularis* var. *lissosperma*, *Hakea acicularis* var. *smithii*, *Hakea obliqua*, *Hakea decurrens*, *Hakea sericea* var. *lissosperma*, *Hakea tenuiflora*, *Hakea tenuifolia*, *Hakea tenuifolia* var. *typica*) [1] is a shrub or small tree of significant invasive potential that represents a serious threat to biodiversity in several ecosystems outside its native range [2,3,4,5]. *H. sericea* invasions reduce species richness, increase fire risk, change fire regimes and fuel loads, and reduce water run-off, among many other effects [6]. Anthropogenic modifications of landscape and ecosystem structure and processes [7,8] contribute to further increasing the geographic scale of the biological invasion processes, impacts and responses over time [9]. *H. sericea* has become invasive in several regions and countries and is one of the 622 most dangerous invasive species globally [8]. *H. sericea* has been recognized as invasive in South Africa (Fynbos biome, Cape floristic region) since 1925 and has invaded ecosystems in Angola, New Zealand and several European countries [10]. In Europe, it became listed as one of the 16 preferred species for control in Europe [11] after invading areas in France, Spain, Madeira Island (Portugal) and Portugal mainland, where, in 1930, it was introduced for living fences and ornamental purposes [12]. It has also been listed on the EPPO (European and Mediterranean Plant Protection Organization) Alert List since 2007, considered invasive since 2012, with priority for risk assessment since 2016, and a potential future risk in Israel (Israel’s least wanted Alien Ornamental Plant Species) [6,12]. *H. sericea* is on the “A2 List of pests recommended for regulation as quarantine pests” and in the “14 dynamic sheets of invasive alien plants in the EPPO global database”. In France it is classified as invasive [13] and in Spain it is included in the Royal Decree 1168/2011 (list of potential invasive species). In Portugal it is part of the National List of Invasive Species [14].

The *Hakea* genus includes 159 sclerophyllous species with high variation in leaf and flower forms, with 35 million years of evolutive history resulting in high genetic variation [15]. *Hakea* originated in southwestern Australia and gradually spread across Australia, adapting to nutrient-poor environments [16]. Currently, *H. sericea* occurs naturally in dry sclerophyllous forests and heats in coastal regions of southeast Queensland to southeastern South Wales, Australia. It is characterized by terete leaves (flat, rigid leaves), small yellow-to-white flowers in axillary clustered umbelliform inflorescences on short rachises and woody fruits (follicles). It is a warm temperate climate shrub or small tree with a high drought resistance and a wide temperature amplitude and habitat range [17]. *Hakea* serotinous fruits are of adaptive significance in fire-prone ecosystems [18,19] since they are resistant to heat and herbivory and, even after the death of the plant, continue to release seeds for long periods of time, building up a large seed bank [18,20,21]. The genus *Hakea* has proteoid (or clustered) roots that exert a great influence on the use of resources. These consist of root hairs clustered in longitudinal columns, which increase the contact surface by up to 300 times, and exude carboxylates, phenols, phosphatases, and organic acids, increasing the solubilization of mineral and organic nutrients and the absorption of P complexed with Ca, Fe, and Al [22,23,24].

To technically and scientifically support the definition of management, monitoring and control measures of *H. sericea* and the development of public policies to mitigate impacts requires a thorough review of the published research. This is even more urgent in recent areas of expansion of *H. sericea*, such as the Iberian Peninsula [11], but also in areas of potential expansion in the future. Furthermore, it is necessary to systematize the scientific knowledge on particular fields, such as ecophysiological, morphological and evolutionary adaptive mechanisms, in its center of origin and in invaded ecosystems, as well as on the assessment of local and regional impacts of invasions. The systematization and synthesis of experimental and empirical data concerning the relationship between species autecology and fire contributes to provide answers to research questions [25]. Systematic reviews are an important tool to evaluate and use information relative to plant invasions and their control methods [26,27]. Pullin and Knight, Pullin et al., and Sutherland et al. argue that the systematic review methodology improves the identification and provision of evidence to support practice and policy in conservation and environmental management [28,29,30].

The objective of this paper is to synthesize the scientific knowledge about *H. sericea*, covering fields relevant from a biological invasion perspective, such as ecology, ecophysiology, adaptive evolutionary mechanisms, interactions with fire, and the effectiveness of the control management and monitoring of invasions. Although other literature reviews have been conducted to address particular issues related to *Hakea* species invasions (e.g., [23,31,32,33]), this is the only systematic literature review simultaneously covering the aspects that are relevant to understand *H. sericea* as a very successful invasive alien species and its control in Mediterranean regions. The knowledge gathered in this review can be of use for decision makers and stakeholders to support *H. sericea* invasion management, monitoring and control programs and measures in the Iberian Peninsula and elsewhere, and effective public policies to prevent or minimize impacts.

## 2. Results

The search performed in three databases returned 985 publications (Table 1). Of these, 339 were kept after removing repetitions, 284 after removing publications related to analytical chemistry, toxicology, biochemistry and zoology of other Proteaceae species, and 207 after removing publications not related to the three formulated questions (Figure 1).

The majority of the publications were related to questions 1 (47.4%) and 3 (26.1%), followed by publications related to question 2 (11.6%) and the associations of questions (5.8, 6.3 and 2.9% for combinations 1 and 2, 1 and 3, and 1, 2 and 3, respectively) (Table 2). Regarding date, although the first work was published in 1953, it was only after 1982 that publications became regular. An increasing number of publications was observed between 2014 and 2021 (Figure 2). The full list of publications selected in the systematic literature research can be accessed in Appendix A.

The applied search strings and the following review process (Figure 2) returned 114 publications (55.1%) specifically related to *H. sericea*, either as the studied species or included in the database used for analysis (Table 3). Other publications were more general, referring to a large number of *Hakea* species (Table 3).

Australian and South African researchers dominate the authorship of publications (Table 4). The work of Australian researchers, authoring 48.8% of the selected publications, was mainly related to questions 1 (66.3%) and 2 (15.8%). South African researchers authored 25.6% of the selected publications, mainly related to question 3 (73.6%). Portuguese researchers produced 12 (5.8%) publications, 50% of which related to question 3. Six publications were authored by New Zealand researchers, mainly on topics related to question 2, and nine publications were carried out on a global scale (authors from several countries) (Table 4).

Four publications were authored by South African and Australian researchers and 26 (12.6%) by researchers from Spain, France, the Netherlands, Germany, Chile, Slovakia, United States of America, European Union (as a whole), United Kingdom, Switzerland, or associations of these countries (Appendix A).

The most represented authors/co-authors were from Australian research institutes and universities. Byron B. Lamont (Curtin University), Hans Lambers (University of Western Australia), Philip K. Groom (Curtin University) and Michael Willian Shane (University of Western Australia) were the authors contributing the most to the selected literature on *Hakea*. Considering just research published on Iberian Mediterranean ecosystems, there was a quantitative chronological increase in publications with authorship/co-authorship of Portuguese researchers, in particular Joaquim S. Silva (Polytechnic Institute of Coimbra), Manuel Fernando Souza (University of Minho), Generosa Teixeira (University of Lisbon), Maria C. Morais and João A. Cabral (University of Trás-os-Montes and Alto Douro), Elisabete Marchante (University of Coimbra), Hélia Marchante (Polytechnic Institute of Coimbra), João Martins and João P. Honrado (University of Porto), and Cláudio Araújo-Paredes (Polytechnic Institute of Viana do Castelo).

Publications with authorship or co-authorship of Australian researchers were mostly related to autecology (e.g., [34]), reproductive biology/ecology (e.g., [35,36]), phylogenetics (e.g., [37]), evolutionary biology (e.g., [16]), biogeography (e.g., [38,39]), functional (e.g., [40]), physiological (e.g., [23]) and morphological characteristics (e.g., [41,42]), and responses to biotic and abiotic determinants (e.g., [43,44]) of *Hakea* and *H. sericea*, all related to question 1.

Publications with authors or co-authors affiliated with South African research institutions (the first country to scientifically document a biological invasion by *H. sericea*) predominantly addressed control techniques (and their history) of *H. sericea* (question 3) in South African ecosystems, especially in the Fynbos region (e.g., [4,45,46,47,48,49,50,51,52,53,54]). Other *H. sericea*-invaded countries showed a significant proportion of publications related to question 3, such as Portugal (50%) and New Zealand (16.7%)

It was observed that 11.5% (*n* = 25) of all publications selected were related to responses of the genus *Hakea* to fire (question 2). Other publications addressed fire in combination with topics in other questions, namely in question 1 (5.8%) and questions 1 and 3 (2.9%).

## 3. Discussion

### 3.1. Ecological and Environmental Factors Contributing to H. sericea Invasiveness

Almost all the autecology publications selected referring to research conducted in Australian ecosystems emphasized adaptations that allow *H. sericea* to be a successful competitor in its native range and in introduced ecosystems [4,55]. Heat, cold and drought conditions in dystrophic environments do not appear to represent environmental obstacles for *Hakea* species [24]. There are a series of adaptive mechanisms, functional traits and integration functions developed thorough the evolutionary history of *Hakea* genus that drive survival and reproduction in a broad ecological spectrum and explain to a large degree the success of many of its species [56,57]. In the case of *H. sericea*, the invasiveness of the species in Mediterranean ecosystems relies on characteristics that improve its competitive advantage potential over resident species [10]. These characteristics of invasive plants are further enhanced under climatic change conditions, which increase the invasion impacts [58,59].

#### 3.1.1. Evolutionary History

The 35-million-year evolutionary history of the *Hakea* genus has allowed for wide genetic and phenotypic variation with significant contributions to increasing competitive ability outside its native range [15]. Through the influence of evolutionary competition, the high diversification and the high invasion potential of the competition mechanisms found today in *Hakea* seem to be density-dependent and to increase with the number of coexisting species [15].

Based on a phylogenetic analysis of *Hakea* taxa (135 species and 146 species and subspecies), [60] argues that the biodiversity of Mediterranean climate ecosystems is usually explained as resulting from a large ecological carrying capacity with large diversity accumulation over time relative to other world ecosystems. The *Hakea* genus originated in Australian southwestern sandy plains and speciated and migrated to forest and dry granite environments around 12 MYA (million years ago), adapting to nutrient-poor ecosystems [16]. Biome evolutive changes are linked to key physiologic mechanisms for lineage adaptation in new environments. *Hakea* species have widespread biogeographic ranges, and their distribution today occupies nearly 60% of the Mediterranean Australian biomes in the southwest and southeast regions. The speciation variation in *Hakea* is dynamic in the sense that time and opportunity limitations for dispersal outside of its ancestry areas have less of an influence on the shape of distributions of current species [39].

The *Hakea* genus ancestry can be described from fossil fruits from the mid-Tertiary period associated with tropical pluvial and monsoon forests [61]. The very efficient adaptative strategy of the *Hakea* genus for wide light variations can be explained by the condition of displaying amphistomatic leaves (stomata on both leaf faces). This can be associated with open vegetation areas, suggesting a strong evolutionary convergence to high water transport and gas exchange efficiency, with enhanced photosynthetic capacities, which may be one of the most important strategies for a highly competitive status in relation to resident species [62].

#### 3.1.2. Reproductive Strategies, Water Use Efficiency and Drought Resistance

There have been complex ecological interactions between *H. sericea* and resident species involving reproductive and resource use capacity during the naturalization and invasion stages in different biogeographic areas, such as Africa, Oceania, and Europe [7]. *H. sericea* propagule pressure has been recognized as a positive factor in invasion establishment [4]. Additionally, human activities and land use change enhance the spread of *H. sericea*, with agriculture being the main cause of these changes, in particular when management is neglected [63]. Successful invasive species show superior fitness at least in one growth variable related to light, nutrient, or water use [7]. Soil chemical properties are recognized as ecological drivers in invaded areas, and lower nutritional and water requirements can contribute to the spatial expansion of plants adapted to dry, low-fertility soils in climate change scenarios, reducing the distribution of species restricted to a specific soil condition [64]. Ding et al. observed a general decrease in woody species occurrence and plant richness patterns with higher P availability in Australian ecosystems, although this effect was balanced by aridity [64].

*Hakea* is considered the world’s most sclerophyllous genus [16]. Limited water availability in high-luminosity ecosystems induces sclerophylly in *Hakea* species, linking leaf morphological characteristics with water use efficiency [41]. The high phenotypic variability of the *Hakea* genus comprises two leaf types, broad and terete (acicular like) leaves. *H. sericea* terete leaves are more sclerophyllous than broad leaves, have higher mass per area unit, increase with plant age [41], and have lower transpiration rates, which is an adaptation mechanism to avoid hydric stress known as drought avoidance [42].

*Hakea* adaptive mechanisms operate in a variable way and will determine the tradeoffs between above- and below-ground resource allocation [65]. This results from the phenotypic plasticity of *Hakea* species with terete leaves, but also from endogamy avoidance caused by a self-pollination impediment, ensuring genetic variability within populations, thus contributing to the persistence of genetically isolated populations prone to fluctuations in their size [66]. The phenotypic plasticity of *Hakea salicifolia* in invaded areas allows for the establishment of interactions with resident pollinators (in the absence of its native pollinators), enhancing fruit production [67].

In addition to *Hakea* adaptation mechanisms, there is a combination of life forms (resprouters and non-resprouters) and morphological characteristics of leaves in response to fire that need to be taken into account. The division between resprouters and non-resprouters and terete and broadleaf species is related to recruitment, fire frequency, temperature and precipitation. In the literature, *H. sericea* is classified as a non-resprouter. There is, however, evidence that the species may resprout when trunks are cut too high, although that behavior is not frequent [68]. Terete leaf species and terete leaves are related to drier areas of low to moderate precipitation rates and non-resprouter species to low annual temperatures [38]. The ecophysiological mechanisms of *Hakea* species depend on leaf morphology, life form, and habitat specificity, and this dependence is prominent under high temperatures [69]. Different leaf types and resprouter and non-resprouter species will differ in their response to water stress. For example, low water availability has led to specific phenotypic plasticity and adaptation to arid conditions, such as lower specific leaf area and high resistance to xylem cavitation in 16 *Hakea* species [70]. On the other hand, invasive species with smooth stems and broad leaves can increase stem flow, increasing the amount of water reaching the soil [71].

*Hakea* species with terete leaves have higher Huber values (e.g., sapwood cross-section divided by distal-to-stem leaf area) and employ a strategy to prevent hydric stress, reducing plant exposure during high-evaporative-demand periods [72]. *Hakea* sp. access sparse rainwater with shallow roots, making them less vulnerable to cavitation, mediated by a high implosion resistance from conductive vessels. The sapwood density also influences water use, and *Hakea* sp. present a small conductive xylem area with high branch and wood densities, with low hydraulic conductivity potential [72]. In Portuguese ecosystems, *H. sericea* has shown seasonal variations in water use and photosynthetic parameters, with high water use efficiency throughout the year, maximal stomatal conductance in the spring and maximal photosynthetic rates in the fall [5].

#### 3.1.3. Seeds, Follicles and Serotiny

Depending on environmental and soil characteristics, the germination of *H. sericea* seeds may start after 30 days, peaking for a few months after their release from follicles [35]. When seeds are protected by their woody fruits, germination viability can be higher than 80% [34,73]. However, some Australian serotinous species such as *H. lissocarpa* are vulnerable to thermal shock, decreasing germination rates when exposed to a temperature of 100 °C [74]. Within the *Hakea* genus, the variation in seed size is high (2 to 500 mg) and seed size is negatively related to fecundity [75]. Non-resprouter *Hakea* species produce a higher number of seeds with less variation in seed size and mass than resprouter species [75]. On the other hand, *Hakea* species with strong serotiny (seed retention in the follicles up to 10 years [75]) tend to invest more energy in heavier follicles with thicker walls, increasing the follicle–seed weight ratio, with no difference in seed weight [76]. The variation in seed and follicle size is attributed to restrictions of pre-dispersal adaptive traits to avoid losses caused by seed predators and fire [77]. In *H. sericea*, seed mass is positively related to follicle mass [77]. Bigger seeds generate seedlings dependent on cotyledonary reserves, from which N, P, Ca, and Cu have been translocated [36]. In addition, *Hakea* seeds have strong resistance to mechanical damage [78]. In general, big seeds in the *Hakea* genus show lower physical and chemical stress than small seeds [79].

Resistance to dystrophic environments is facilitated by seed organic reserves (which contain no starch) [80]. In this sense, high N- and P-specific storage by *H. sericea* seeds (even in unbalanced amounts in relation to other nutrients), in combination with high dispersal capacity over long distances and high adaptability under dystrophic conditions, represents an important competitive advantage over resident species [81]. P levels in *Hakea* sp. seeds are 300–500 times and N, Cu, and Zn levels are 8–100 times higher than in most species [81]. Seeds, representing 1–16% of the total fruit dry mass, contain 83–98% and 43–78% of P and N fruit content, respectively [81]. *H. sericea* seeds have mechanisms of N and P remobilization that, together with water retention mechanisms, ensure germination in arid and dystrophic conditions [82]. P seed concentration in *H. sericea* is higher than in other Proteaceae and is found mainly in phytic acid, phospholipids, RNA, DNA, phosphorated proteins, polyphosphate and orthophosphate forms [83]. In a review of P accumulation in Proteaceae seeds, ref. [84] reported that *H. sericea* seeds present higher levels of P content (9.9 and 11.7 mg·g^−1^) under invasion conditions. These values are less than those found in *H. sericea* in Australian ecosystems [84]. Authors also found that less P-limited ecosystems led to less seed P storage and that P content was directly related to serotiny levels, with higher serotiny found in dystrophic soils [84]. *H. sericea* phenotypic plasticity allows the production of smaller seeds in less P-limited ecosystems [84]. In Australian ecosystems, 70% to 80% of *H. sericea* follicle P content is invested in seeds, which is almost five-fold higher than that observed in *H. sericea* plants in South African ecosystems [84].

*H. sericea* high seed production, large canopy seed bank, seed dispersal syndrome, dispersion over long distances and the adaptations to survive under recurrent fires are important factors of plant establishment. In general, invasive alien species have a higher probability of establishing long-distance dispersal [35,85,86]. In the case of *H. sericea*, anemochory is one of the most important determinants of post-fire seedling establishment [87]. Groom, analyzing dispersal in 14 *Hakea* species from west Australia, found that seeds were generally dispersed over 20 m from the nearest plant, with larger winged seed surface–mass ratios generating greater seed dispersal terminal velocities [87]. Among Proteaceae species, *H. sericea* winged seeds show the highest seed dispersal distance, with high stability in strong wind situations, allowing the colonization of new areas far from original populations [87].

*H. sericea* has high seed longevity inside fire-resistant follicles and a high potential to form dense canopy seed banks [88,89]. This capacity is further enhanced in the absence of natural consumers [89].

Plant fire resistance is not a key mechanism involved in *H. sericea* population establishment, but follicle serotiny seems to be determinant for the continuity of its populations [89]. Follicle serotiny is a strategy of storing seeds in the plant canopy, releasing them in response to an environmental trigger, often fire [21]. High serotiny levels are linked to dystrophic soils and to seed P content [84]. Serotiny can take place even after the death of the plant [18]. The *Hakea* genus presents the third highest serotiny index among the 52 genera analyzed by Lamont et al. [90]. Although advantageous for the successful plant establishment after disturbances, serotiny involves high maintenance costs [76]. Strong serotiny (as found in *H. sericea*) allows seed retention up to 10 years, but it increases photosynthetic and transpiration rates for pre-dispersal seed protection using large and dense follicles with thick walls to avoid seed predators [76]. In a review, Lamont et al. argued that fire acts as a selective agent of serotiny levels in Mediterranean ecosystem plants, and recent changes in climate and land use have had a strong impact on the conservation status and evolutionary trajectory of serotinous plants in Mediterranean ecosystems [90]. Serotiny confers adaptive benefits when fire occurs between the time of reproductive maturity and the plant life span, and this fact must be considered when planning the control of species with prescribed fires. *H. sericea* annual fecundity rates and the number of flowering events after fire are related to serotiny levels, which result from fruit size and plant architecture [77]. Due to the anisotropic retraction of the follicular valves, the follicles can open after fire or after being separated from the plant [91].

#### 3.1.4. Seedling Establishment

*H. sericea* seedlings show a high rusticity and resistance to environmental biotic and abiotic conditions, and their growth is not affected by herbivory or the presence of pathogens [44]. *Hakea* seeds in general have high P, Fe and protein content and do not require high-moisture conditions for germination [82]. Seedlings contain a high concentration of chlorophyll, soluble sugars, and have high photosynthetic rates and water, P and N use efficiency [24,92,93]. In *Hakea lasianthoides* and *Hakea psilorryncha*, seedlings under high-light and low-water conditions, the high cotyledon nutrient content maximizes the carbon supply for fast foliar expansion and contributes to the nutrient supply for seedlings [94]. Additionally, P uptake increases with P supply, resulting in an increase in *Hakea preissii* seedling biomass [95].

*H. sericea* seedling establishment and growth in invaded areas are enhanced by a lack of natural enemies, pathogens, and specialized herbivores [96,97]. Nonetheless, 14 *Hakea* species in Australia showed high concentrations of phenolic compounds to protect against large herbivores, such as kangaroos [96,97]. The lack of resistance to waterlogging conditions, with the formation of adventitious roots near the soil surface, can negatively affect the seedlings of *Hakea* species [98].

#### 3.1.5. Proteoid Roots, Phosphorus and Nutrient Use Efficiency

Proteoid roots (or clustered roots) consist of an alteration of plant root hairs with the formation of closely spaced lateral roots at a density of about 10–1000 radicels per centimeter of the parent root [99], increasing the root contact surface by 140 to 300 times [100]. Proteoid roots are associated with the uptake of low-mobility nutrients in eight plant families and are linked to P and micronutrient acquisition [101,102,103].

Proteaceae species are strong competitors in dystrophic environments, to a great extent due to their proteoid roots that exude carboxylates, citrate, malate, malonate, lactate, acetate, maleate, fumarate, cis- and trans-aconitate, organic acids and enzymes into the rhizosphere [104]. P uptake is increased by the solubilization of aluminum and iron phosphates, solubilizing absorbed P in soil particles and P complexed with Ca, Fe and Al [23,102,104], with significant mucilage release [105]. P is allocated to photosynthetic mesophyll cells rather than to epidermal bundle sheath cells, resulting in a higher P concentration in seeds and a lower P content in leaves [24]. Jeffrey described a mechanism mediating this P photosynthetic efficiency that consists of the use of a metabolic P route through polyphosphate synthesis and accumulation, with a large proportion of labile phosphate detected in root exudates [106].

Soil chemical and physical properties are important factors in proteoid root formation. The vertical distance of the root from the soil surface, soil density and P deficiency positively influence proteoid root formation [105,107,108], whereas high levels of soil organic matter and N negatively influence proteoid root formation [107]. In Mediterranean conditions, a progressive water supply leads to the development of proteoid roots, decreasing the above-soil field capacity. Proteoid root formation occurs in winter–spring, and dormant roots can be formed in the summer if there is enough moisture in the root system [100].

The low N and P availability in Portuguese soils in invaded areas has led to an increase in *Hakea* proteoid root formation, with dense proteoid roots near the soil surface [101]. The low affinity of inorganic P transport is halved due to the high inorganic P availability, increasing P toxicity in response to increased P tissue concentration, due to an inability to downregulate P uptake, even in concentrations well below those observed for other species [93,101,109,110]. Shane et al. and Shane et al. described an increasing P toxicity with increasing Ca supply due to leaf P displacement into mesophyll cells [111,112], increasing tissue P concentrations [113].

N and P use and the NP ratio in plant tissues are important characteristics of plant ecology that are related to plant distribution and plant invasion processes across ecosystems. The observation of NP ratios in plant tissues worldwide can generate insights into biogeographical and biogeochemical processes involved in plant community dynamics [114,115]. The invasive success of *H. sericea* in South Africa and its high dominance in Australian ecosystems can be attributed, in part, to environmental filters for adapted lineages for these ecosystems via stoichiometric NP ratio homeostasis, supported by a conservative N and P use mechanism [115,116,117]. Invaded Iberian ecosystems present NP ratios (9.9) similar to those found in Australian ecosystems (11.2) [115].

Globally, the successful Proteaceae establishment in low-P environments is linked to lower phospholipid (−37%) and P contents (−50%) in mature than in young leaves. Thus, there is lower P resource investment in phospholipids than in non-phospholipids, without photosynthesis prejudice, and this difference is higher than that observed for species of other families [92]. Lambers et al. pointed out that there is an indirect relationship between P use efficiency and Mn in the nutrition of 727 Australian and New Zealand species [103]. Proteoid roots of *Hakea* species have enhanced performance under high soil Mn concentrations, associated with enhanced root carboxylate production, leading to the highest NP ratios among 747 plant species (between 32.3 and 53.3) from several world biomes [116]. This is related to the tight control of nutrient absorption in low-P environments [116,117,118,119].

In high-P-availability conditions, proteoid and mycorrhizal rooted plants compete for P acquisition. Inversely, in low-P environments there is a biotic relationship of P facilitation, where proteoid roots benefit from P immobilization by mycorrhizal roots [109]. As soil ages, the allocation of P to inorganic P, phospholipids, and nucleic acids decreases, while N remains constant. Thus, highly weathered soils tend to exhibit higher NP ratios in plants, even in non-P-limited environments [120]. This mechanism includes high P resorption rates (up to 85%), increased N and P photosynthetic use efficiency, and fast photosynthetic rates per unity of P with increased soil aging [94].

#### 3.1.6. Synthesis

The main characteristics of *Hakea* species (including *H. sericea*) related to the first formulated question (“What ecological and environmental factors (biotic and abiotic, morphological and ecophysiological aspects of the genus and species) contribute to the establishment of *H. sericea* populations?”) are synthesized in Table 5.

### 3.2. Fire and H. sericea Invasions

Fire has a significant effect on ecosystem dynamics and species evolution in fire- and non-fire-prone ecosystems. The coexistence of plant communities and fire has shaped many plant traits such as fire-resistant meristematic tissues, post-fire regrowth and basal resprout, seed release and germination from dead plants from persistent seed banks, and post-fire floral stimulation [121]. From an evolutionary point of view, plant traits in response to fire evolved slowly in the Cretaceous and intensely in the Cenozoic periods [121]. *H. sericea* has developed mechanisms to avoid and respond to fire that constitute key traits for its invasive success in Mediterranean regions. In a revision paper of fire seasonality effects on plant populations, ref. [122] stated that there are several mechanisms of post-fire population recovery and that *H. sericea* employs them all: adult survival and growth, propagule availability and seed production; seed bank availability; juvenile growth; high seed temperature tolerance; post-fire seed survival; and post-fire seedling establishment.

#### 3.2.1. Hakea Post-Fire Responses and Strategies

Fire positively contributes to the invasive potential of some plants, promoting biological invasions and threating biodiversity, mainly by changing the structure of the community and ecosystem and the fire regime [9,123,124,125]. Responses to different fire intensities can be quite variable in different plant species [126]. In Portugal, wildfires and plant invasions can establish positive feedbacks with impacts on fire regimes [127]. Fire facilitates invasions due to adaptive fire traits of invasive plant species (including *H. sericea*) that make them able to coexist with, or even take advantage of, fire [127]. Some exotic species have their competitive and invasive capacity enhanced by increased recruitment in burned areas, especially if fire is followed by rainfall. Mountainous ecosystems in Portugal frequently subjected to forest fires have been subjected to the well-established biological invasion processes of a set of exotic species, with *H. sericea* being one of the most recent cases [128].

In Portugal, *H. sericea* presents better performance than resident species, such as *Pinus pinaster* [5], increasing in abundance after intense fires [129]. Currently, dense (>75% cover) stands of *H. sericea* negatively affect several ecosystem functions and services, such as recreation and animal production, due to difficulty or impossibility of access to humans and wild and domesticated animals [5,12,130].

Fire resistance in 82 *Hakea* species is linked with a regeneration strategy [75]. Regeneration strategies observed in *Hakea* species in fire-prone ecosystems, namely resprouting and obligate seeding (such as in *H. sericea*), can explain their wide distribution in fire-prone and sclerophyllous ecosystems in poorer and drier soils [131]. The resource allocation patterns in obligate seeders differ from those in resprouter species and are based on seed production investments to enhance the probability of survival after wildfires [131]. Resprouters respond to fire as individuals and obligate seeders as populations. Resprouters regrow after fire by activating dormant vegetative buds. Obligate seeders are killed by fire and re-established through the germination of the seed bank or by serotinous follicles [131]. Overall, approximately 52% of *Hakea* species are resprouters and 48% are obligate seeders [132]. Ecosystems with large proportions of resprouters are affected by more intense and stressful fire regimes [132]. However, seed production represents a crucial mechanism for population establishment and survival in serotinous Australian plant species. That is the case of *H. sericea*.

Resprouter *Hakea* species tend to produce low-viability seeds [34]. The seeds of *Hakea* species that invest in a seed bank tend to be stimulated to germinate after being subjected to high temperatures [34]. *H. sericea* follicles characterized by lignification and an opening on both sides (unique in Proteaceae) are very resilient fruits, responding to fire and separation from the plant by means of anisotropic retraction of the follicle valves [91].

However, *H. sericea* seeds, on the plant and on the soil surface, do not tolerate heat above 100 °C. Pepo et al. reported that *H. sericea* follicles heated to a temperature of 120 °C and seeds heated to 60 °C decreased seed viability by 90% [133]. These authors also found that after fire occurrence, buried seeds were more likely to survive than seeds on the soil surface and inside the follicles [133]. Bradstock et al. argued that *H. sericea* follicles protect seeds from high temperatures through the dimensions of the fruit (8.9 mm abaxial wall, 14.8 mm between seed and follicle wall, 2.1 mm superior wall, 14.5 mm lateral wall, and 6.4 g of weight) [20].

Additionally, in *H. sericea*, anemochoric dispersal facilitates plant establishment in hot areas over relatively long distances (around 12 m away from the parental stand) [86]. In this case, the *H. sericea* post-fire seed bank is strategic in granting high recruitment efficiency rates in Australian ecosystems, which is partially dependent on the follicle structure to protect seeds from heat released in wildfires [20].

Resource allocation in post-fire serotinous species depends on environmental factors and the surrounding communities [134,135]. In invaded New Zealand ecosystems, a high fire frequency was associated with enhanced bush dominance and an enhanced frequency of alien species, favoring serotinous species that disperse rapidly after fire, affecting natural succession [136]. As a result of this pattern, *H. sericea* dominates drier and frequently burned areas with low nutrient availability in New Zealand, where its plant cover is inversely proportional to the cover, height and richness of native plant communities [137]. Under these conditions, fire intervals of up to 10–20 years are also related to decreased species richness at the landscape scale in New Zealand. For this reasons, fire intervals of more than 15 years must be avoided [136].

The coexistence of *H. sericea* with fire and ecosystem management in New Zealand is very complex and site-specific. According to Clarkson et al., Gumland ecosystems are critically threatened by *H. sericea*, which occupies 45% of the canopy cover and is present in more than 50% of the plots these authors analyzed [138]. Frequent burning is not carried out in this area as it leads to biodiversity loss and increases the susceptibility to invading fire-adapted plants [138].

#### 3.2.2. Seed Bank and Post-Fire Recruitment

Different fire intervals adversely affect dry sclerophyllous plant populations in Australian temperate ecosystems [139]. For *H. sericea,* seed germination after fires may vary widely between areas, depending on fire intensity and fire periodicity [140], resulting in variations in seed bank dynamics over time. In New Zealand ecosystems, *H. sericea* showed increasing reproductive ability in post-fire areas, reaching 260 follicles per square meter [88].

Ecosystem flammability, ignitability and combustibility are driven by the dominant species in plant communities in New Zealand. Recurrent fires benefit *H. sericea* by excluding resident species in regenerative stages, changing successional patterns. Responses of *H. sericea* to frequent fires can lead to its dominance, thereby altering an ecosystem’s fire characteristics [141]. After 34 years of frequent prescribed burns in Australian ecosystems, *H. sericea* follicle production per plant remained high [141]. The authors observed low individual mortality and follicle production increased with the increasing height of plants. This could be an *H. sericea* evolutionary response to frequent burns due to the selection of highly reproductive individuals and populations [142].

In Australian ecosystems, spring burns resulted in high recruitment levels in relation to fall burns, accompanied by the overcoming of seed dormancy. This pattern is more prominent in seasonal ecosystems, where the regulation of post-fire recruitment patterns occurs due to combinations among optimal temperature for seed release, seed germination, seedling establishment and growth, pluviosity and interactions with the plant community, predators and herbivores [143].

The fire season does not necessarily increase germinable seed release after a fire [134]. Prescribed burning in a single season may cause composition changes and diversity decline by only affecting particular species [89].

#### 3.2.3. Fire Modeling of *Hakea* Invasions

There is an increasing trend of publications on the modeling of fire and its relationship with biological invasions. Publications on the interaction of fire and *H. sericea* invasions often refer to ecosystems where prescribed fire is a relevant management tool, such as in Portugal and South Africa [130,135].

In Iberian Mediterranean ecosystems, changes in fire behavior in response to the invasion of *H. sericea* have been observed and modeled. In central Portugal, *H. sericea* stands showed the highest flammability and fuel load among fuel models developed for shrub communities, especially in fine dead fuels that contribute to initial fire. *H. sericea* total fuel biomass was higher than estimated, on average, in models of local Atlantic shrub communities, also showing a high fuel depth derived from litter production [34]. The expansion of plant communities dominated by *H. sericea* significantly increases the probability of changes in the structure of communities, establishing positive feedback with fire behavior patterns, further catalyzing community changes [130].

Likewise, the modeling and simulation of fire behavior in areas invaded by *H. sericea* in South African Fynbos under different fuel scenarios indicated that the invasion considerably alters native communities due to a 60% increase in fuel load and a 45% decrease in leaf moisture, leading to an increase in frequency, intensity and fire hazard [144]. In fire regime modeling in Fynbos communities, Tonnabel et al. pointed out that resource allocation patterns in serotinous obligate seeder species increase their survival probability in short fire intervals [135]. Strong serotiny is an optimal strategy between fire events, when a seedling is exposed to high competition with adult individuals [135].

#### 3.2.4. Synthesis

A synthesis of traits and effects involved in responses to fires in *H. sericea* populations is presented in Table 6.

### 3.3. Control Methods, Monitoring and Management of H. sericea Invasions

Management towards the control of *Hakea* and other invasive species is a complex and difficult process in countries dominated by fire-prone ecosystems [35,145]. This is reflected in the growing costs of control of *H. sericea* that might have an impact on local and national economies. In the South African Cape floristic region, an area strongly affected by invasive alien species, including *H. sericea*, historically, control costs have amounted to around 78.7 million USD per year and it is estimated that they will be in the order of 92 million USD per year in the near future [145,146]. In only the last 20 years (1996–2016), 38 million USD were spent, 90% of which were for the control of *Acacia*, *Pinus* and *Hakea* species. The cost to keep invasions under control is estimated to be between 1.3 and 174 million dollars [146,147]. Although major efforts were undertaken to try to control *H. sericea* [51], the most important alien invasive species in the Fynbos biome [148], the results were variable and invasions of *Hakea* species could not be effectively controlled. However, knowledge and expertise in control, monitoring, and management methods have evolved significantly [32,46,51,149]. More recently, New Zealand and Portugal have conducted research on the control, monitoring and management of *H. sericea* populations impacting the scientific community [11,127,129,130,150,151,152]. In this section, we structured the main findings of the literature review on control methods (mechanical and prescribed fire, chemical, biological and integrated control management), modeling and monitoring, and the management and of *H. sericea* invasions.

#### 3.3.1. Mechanical Control and Prescribed Fire

Mechanical control is globally the oldest method of weed control and is extensively used in agriculture and environmental management [153]. In South Africa, early *H. sericea* control records report the use of mechanical clearing by felling, followed by prescribed fire [51]. This management technique has been used massively from the mid-1970s and into the following two decades [51]. For this, the mountain areas between Cape Town and Port Elizabeth were divided into blocks burned in 12-year cycles [51]. Currently, mechanical *H. sericea* plant removal (clearing), with or without prescribed fire, is a common and promising technique to reduce the density and extent of infestations when human and machinery access is possible [51]. Operations must consider, however, their financial costs and plant phenological patterns to control seedlings that germinate from the seeds of dead plant follicles [154].

Prescribed fire is also a common technique applied in the control of *Hakea* invasions, often combined with mechanical control [50]. In South African Fynbos, Richardson and Van Wilgen did not observe an effect of the seed release season following felling on *H. sericea* regeneration in burned and unburned plots [154]. They observed that the use of intense fires reaching *H. sericea* juveniles and seedlings was more promising than manipulating seedling survival in relation to the felling season [154]. In contrast, *H. sericea* burnt without previous felling may be a management alternative if fire is applied immediately after seed release [155]. The burning of *H. sericea* dry felled biomass should not be performed due to increased soil fire intensity leading to soil erosion and water and biodiversity losses [155]. Additionally, the application of fire in dense *H. sericea* stands amplifies ecosystem impacts. Since fire intensity in invaded areas tends to be severe, burning mechanically removed *H. sericea* biomass must be performed while taking into account the meteorological conditions in order to avoid and prevent high-intensity fires [155]. In Fynbos ecosystems, 19 months after an accidental fire affected an area with *H. sericea*, only 13% of the cover with native vegetation and 42 plant species richness had recovered [156]. Additionally, in Fynbos ecosystems, it was recommended that slash and burn operations for *H. sericea* control should be planned to reduce intervals between the mechanical removal of newer-generation plants and the established older populations in areas of difficult access [50]. After that, it was recommended to construct terraces following the contour of the land to avoid soil erosion [50].

In Portugal, slashing and burning associated with fire management have shown promising results in the control of *H. sericea* [157]. In the Marão mountains, it was found that when meteorological conditions do not allow the conduction of safe prescribed fires, mechanical plant cutting at the stem base may be an adequate strategy. Monitoring and interventions using prescribed fires, if necessary, can avoid population regeneration and viable seed production [157].

Similarly, Bosch tested and compared the effectiveness of slashing and burning to only burning of *H. sericea* in Portugal, in comparison to no burning, and found no differences among these treatments, both being successful in terms of reducing *H. sericea* densities in the short term to 1 seedling/m^2^ and the relative abundance to less than 3% one year after prescribed fires [158]. Experiments on mechanical cleaning followed by sequential fire management are currently being conducted in the control of *H. sericea* in Portugal to address whether different fire intervals and seasons will impact floristic structure and composition, seed bank properties, and soil erosion [127].

#### 3.3.2. Chemical Control

Weed chemical control is a controversial issue due to the multiple and complex interactions between agrochemicals, nature, and society, and the negative effects observed on the environment and human health [159]. Additionally, increasing herbicide resistance and decreasing herbicide effectiveness over time can impose greater impacts on the environment and costs on society [160].

Herbicide (such as Tebuthiuron, 2 methyl 4 chlorophenoxyacetic acid, 2:4:5 trichlorophenoxyacetic acid) application has proved to be difficult and expensive in the control and management of *H. sericea* populations [161]. Structural, morphological, and functional foliar adaptations, such as a thick cuticle (increasing with plant age) and lower stomatal densities, may explain the higher tolerance of *H. sericea* to herbicide compared to *H. salicifolia* [150]. Teixeira et al. observed lower herbicide (Glyphosate) absorption rates in *H. sericea* that may be associated with leaf cellular water loss strategies such as reduced leaf area, thicker cuticles, and immersed stomata in long-life leaves [150]. The authors recommended the observation of the age and phenological stage of *H. sericea* plants in chemical control planning before the applications [150].

Chemical control as the only method of *Hakea* stand management is usually not recommended due to the high economic costs involved, possible plant resistance, and herbicide lateral movement in the soil impacting non-target species and other biotic ecosystem components [45,48]. Planning and implementing *H. sericea* chemical control must be considered a part of integrated management approaches to reduce costs and maximize the effectiveness of chemical control by combining this with other methods, such as mechanical and biological control [159].

#### 3.3.3. Biological Control

In general, biological control has a slower response time compared to mechanical and chemical control methods and requires previous studies to evaluate potential ecological interactions of biological agents in introduced ecosystems [162]. In addition, for the biological control of Australian native plants, there is a need for evaluating adaptive traits and their competitive interactions [162].

The biological control of *Hakea* spp. has been widely used in South African ecosystems since the late 1960s and early 1970s, using insects introduced from Australia [2,161]. According to Van Wilgen et al., there should be more investment in biological control in South Africa, the use of which has resulted in a decline in *H. sericea* populations, mainly in association with mechanical control [3]. Moran and Hoffmann highlighted the direct and indirect effects of biological control on decreasing the invasiveness of species such as *Hakea sericea* in the Fynbos biome in South Africa, preventing new areas of colonization and decreasing the frequency and intensity of mechanical control and fire operations [54]. According to Moll and Trinder-Smith, the first 30 years of biological control in South Africa reduced the percentage of the occurrence of *H. sericea* through decreasing the abundance of seed production and increasing plant mortality [2].

The first biological control agents introduced in South Africa, in 1970, were *Erytenna consputa* (Coleoptera) and *Carposina autologa* (Lepidoptera). *Cydmaea binolata* (Coleoptera) was introduced in 1979 to suppress seedling recruitment [161]. Later, in 2001 and 2006, other Coleoptera (*Aphanasium australe* and *Dicomada rufa*) were introduced [149]. *C. autologa* was introduced to reduce accumulated *H. sericea* seeds released after fire, but in 1983 the program was suspended due to the better results obtained with mechanical control and the higher effectiveness of *E. consputa* [47]. Additionally, the use of *Colletorichum gloeosporioides* as a fungal biological control agent resulted in the death of *H. sericea* plants, which consequently killed *C. autologa* in all instar stages [47]. *C. autologa* was considered a promising control agent due to its high dispersion and colonization ability, being able to reduce seed accumulation [49]. Between 2012 and 2015, 37.2% of sampled South African *H. sericea* areas contained the insect [163]. Areas of *C. autologa* release must be considered biological control reserve areas [49]. However, Gordon and Lyons observed drawbacks such as high levels of predation by spiders and mites, an inability to distinguish colonized follicles, as well as high neonatal mortality during transport [163].

The use of *E. consputa* in South Africa caused *H. sericea* seed survival to decrease by 36.8% due to the mortality caused by the young insect forms [164]. After 1990, annual *H. sericea* seed production was diminished, especially in areas where accidental burns are less frequent [31]. *C. binolata*, however, did not achieve satisfactory results for *H. sericea* seedling suppression in South Africa [31]. *D. rufa* was tested in Australia and has shown to be *H. sericea* host-specific [31], and does not need the follicle for reproduction, showing fast post-fire population growth rates. In areas in South Africa where it was introduced, *D. rufa* showed fast post-fire *H. sericea* population growth, decreasing fruit production in the early post-fire years [165]. *A. australe* has been released in South Africa for *H. sericea* control since 2001 [166], damaging the basal stem area and roots and causing the mortality of 7–10% of *H. sericea* individuals. This was considered a species of potential interest for biological control because it is specific to *Hakea* species and there is no evidence of migration to agricultural crops [167].

The biological control of *H. sericea* in South Africa has gone through various levels of success affected by the specificities of the biological and ecological biocontrol agents. For example, *E. consputa* has a specific ability for fast post-fire dispersal, *C. autologa* exhibits slow dispersal, and *D. rufa* appears to show difficulties in adapting to the climatic conditions of introduced areas [168].

Despite efforts, large *H. sericea* populations persist in the coastal mountains of South Africa, even after repeated insect introductions. Biological control must be maintained in these areas as new agents are tested and efficacy increases with practice and in combination with other control techniques, as preconized in the integrated management Section 3.3.4. Insect biological control should be considered a priority in these areas as well as the maintenance of *H. sericea*-inoculated stands. These areas must be treated as natural enemy reserves to avoid the extinction and low fecundity rates of biocontrol agents [169].

*Colletotrichum acutatum* (initially classified as *C. gloeosporioides* [170]) was the first pathogen tested and used for biological *H. sericea* control in South Africa. It was isolated from *H. sericea* in South Africa [170], from *Leucospermum* sp. in Portugal [171], and is associated with Proteaceae diseases in South Africa [172]. The fungus began to attract researchers’ attention after the massive death of *H. sericea* stands in South Africa, after which symptoms were monitored and a *C. gloeosporioides* progress matrix was developed to be used in biological control actions [173]. In South Africa, Richardson and Manders observed the high mortality of *H. sericea* caused by *Colletotrichum* sp., with no regeneration of stands close to the infected areas [173]. In initial fungus testing conducted in the 1980s, Dennill et al. reported that the use of *C. gloeosporioides* as a biological control agent was responsible for high mortality in *H. sericea* plants in South African ecosystems [47]. The *H. sericea* gummosis (or anthracnose) disease is caused by *Colletotrichum* sp. and dispersed by infected seeds. Symptoms include dark-colored branches and stems with gummy exudation [172].

In the 1980s, Morris applied an inoculum of *C. gloeosporioides* produced in wheat bran [174]. The application was performed in the cotyledonary phase and resulted in mortality levels of 93% to 98%, but its efficacy was lower in hotter seasons [174]. Based on the promising results obtained from biological *H. sericea* control initiatives until 1999, two bioherbicides were registered in South Africa consisting of a central fragment of gluten wrapped in soybean meal and mycelia of *C. gloeosporioides*, ready to be used [33]. The application can be made via helicopter or by the inoculation of a solution with a water suspension of dried spores, applied to the basal stem part [33].

More recently, another bioherbicide directed for *H. sericea* control was developed in South Africa [170]. The application of this bioherbicide should be repeated to assure the epidemic occurrence of the pathogen for higher effectivity and must be utilized concomitant with the release of *E. consputa* (see insect biological control agents above) [170]. Between 2008 and 2017, the fungus was experimentally applied with variable results. In the first year, the incidence of infected plants ranged from 12 to 95% and the highest mortality rates (38 to 64%) were observed four years after application [170].

Between 2011 and 2020 there was success in inoculating *C. acutatum* via helicopter in inaccessible mountainous areas, and this process was deemed safe for application in *H. sericea* populations in inaccessible terrain [32]. In addition to *Colletotrichum* species, fungi of the family Phyllacoraceae that also cause infections in several *Hakea* species in Australia are options for research on new agents to be used in *Hakea*-invaded areas [175].

Other fungi such as *Pestalotia* spp., *Rhinotrichum* sp., *Cephalosporium* sp., and *Haplosporella* sp. were isolated from *H. sericea* plants in South Africa [33]. With the exception of *Cephalosporium*, no infection symptoms were found [33]. *Phytophtora cinnamomi* (Pythiaceae) caused lesions resulting in high *H. sericea* mortality rates depending, however, on biogeographic aspects and direct solar exposure (conditions that led to the death of conidium) [176]. *Pestalotiopsis funerea* (Amphisphaeriaceae), a fungus causing infections in Proteaceae, was isolated from *H. sericea* plants in Portugal after fire events, showing potential to be used in the biological and/or integrated management of *H. sericea* in this country [177]. In New Zealand ecosystems, Johnston described *Hakea* spp. susceptibility to *Amylostereum sacratum* and *P. cinnamomi* fungi, facilitating future biological control [178].

Besides fungi as bioagents in *Hakea* control, in Australia, a bacterium (*Xylella fastidiosa*) has caused damage and shown high infection potential after artificial inoculation in *Hakea petiolaris* that can be of use in future *H. sericea* control tests [179]. This bacteria, however, is a phytosanitary threat to the diversity of plants of economic interest, which should limit its use in biological *H. sericea* control.

#### 3.3.4. Integrated Control Management

The development of sustainable weed management depends on an integrated system approach that includes herbicides and non-herbicide methods to ensure long-term success. This successful approach can be found in the history of *H. sericea* control in South Africa [3]. A significant decline in *H. sericea* density has been observed in this country due to the combination of biological and mechanical control methods, including a 95% reduction in seed production due to *C. autologa* and *E. consputa*, plant mortality caused by *C. acutatum*, and previous extensive mechanical clearing and burning [54]. Analyzing the effects of control measures also in South Africa between 1979 and 2001, Esler et al. found a reduction of 64% in the distribution of *H. sericea* when mechanical removal was followed by biological control [51]. The integrated management of *H. sericea*, combining the use of mechanical, chemical, and biological control methods and adjusting the intensities of use to specific site and environmental conditions, has therefore shown to be effective in reducing *H. sericea* populations [2,51], eventually allowing control costs and impacts on ecosystem services to be minimized in the future [146].

#### 3.3.5. Modeling and Monitoring of *H. sericea* Invasions

The increasing accessibility to efficient monitoring and modeling tools can assist stakeholders and decision makers worldwide in finding the best approaches to deal with *Hakea* invasions. The monitoring of areas under control programs as well as modeling the spatial and temporal dynamics of invasion factors (including limiting factors) and processes contributes to the understanding and prediction of the spread and trends of invasions, which is of the utmost importance in management [147,180].

In a pioneer study, Richardson performed a cartographic analysis to predict physiographic factors of plant invasion in the Cape region of South Africa, pointing out that quarzitic and arenite substrates are associated with *H. sericea* occurrence and distribution [181]. Also in South Africa, more than 6000 records of *H. sericea* distribution models indicated that rainfall and the mean and minimum temperatures of the coldest month, mountain areas (and areas above 1400 m) with temperatures below 1 °C in the coldest month, sloping regions, valleys, and the presence of cattle pens were the main predictors of the species distribution [52]. On the other hand, a meta-analysis and modeling of floristic data of plant occurrence indicated differences in species composition and abundance with altitudinal variation. Canavan et al. observed that *H. sericea* was exclusively related to lower altitudes (559 registers in low- and 210 in high-mountainous environments) [182]. Modeling also indicated a decrease in *H. sericea* cover and distribution as a result of integrated management, causing a decrease of 95% in seed load, which reduced population growth rates and the formation of new invasive foci [180].

Considering the cost–benefit ratio as a priority decision criterion to manage multiple and conflicting demands, Forsyth et al., using Analytic Hierarchy Process modeling, predicted that *H. sericea* will impact surface water at the landscape scale in South Africa, decreasing runoff by 7 to 56% in a scenario of a maximum occupation of invader species, decreasing ruminant foraging support capacity by 1% [53]. In addition, these authors identified priority watersheds that are not receiving public or private resources and low-priority watersheds receiving significant amounts of resources [53]. Management and conservation models can in this way be put into practice in priority areas and for priority species, assessing cost–benefit when addressing biodiversity conservation and the control of invasive species [147].

Adjacent areas may require different management strategies according to differences in target species, objectives or spatial scales. Considering that *H. sericea* invasion is a transboundary problem for Portuguese and Spanish decision makers, there is a need for mutual efforts and the prioritization of strategic areas for control. Martins et al. used two scales (regional: continental Portugal and Galicia, Spain; and local: northwest of Portugal) for modeling *H. sericea* distribution. The results obtained at the regional scale indicated landscape composition and climate to be the major determinants of the distribution of *H. sericea* over a large environmentally suitable area under consideration, whereas the results at the local scale indicated schist lithology as the main determinant of this distribution [151]. Serralheiro reported that in the Marão forestry intervention zone (Portugal), the main variables involved in *H. sericea* distribution were distance from roads and frost occurrence. Altitude, land cover and insolation were the least contributors to explaining species distribution [157]. Additionally, *H. sericea* mapping based on remotely sensed imagery in northern Portugal resulted in classifications of high accuracy, respectively, of 93% and 75% [152].

Morais et al. developed a *H. sericea* population dynamics model combining several fire risk and control scenarios, differing in periodicity of fire occurrence, control intervention type, and plant cohort age. The results indicated that in the absence of control actions, *H. sericea* populations will continue to expand and that periodic control actions can decrease plant population levels [129].

A growing number of publications on newly available geoprocessing and modeling techniques at various ecological scales will continue to emerge. These tools can support decision making at all levels, providing better quality data on control efficiency and population spread scenarios and on the prediction of financial, social, and environmental cost–benefit ratios, which should be the basis for rational decision making.

#### 3.3.6. Potential Attenuation of *H. sericea* Control Costs

The high costs often involved in *H. sericea* invasion management [147] might limit the application of conventional and potential future control strategies. For this reason, invasive species, such as *Hakea sericea*, might start being seen by stakeholders and decision makers in positive ways, considering the potential advantages of its presence based on the development of alternative uses and value-adding strategies. In Australia, proteoid rooted species, including *prostrata*, have been used in the bioremediation *Hakea* of petroleum-contaminated soils [183]. In this way, the behavior of *H. sericea* also can be tested in order to determine its suitability for the bioremediation of soils contaminated by petroleum or other contaminants. Additionally, phytochemical studies revealed *H. sericea* as a potential source of bioactive compounds [184]. An extract (Alkenylresorcinol) from *H. sericea* collected at Serra da Estrela, Portugal, was demonstrated to have Gram-positive and Gram-negative antibacterial activity for several species, including *Staphylococcus aureus* [184]. *H. sericea* extracts can be used as a food preservative and in alternative therapies for antibiotic use due to its antimicrobial and antibiofilm activities [184]. An aqueous extract from *H. sericea* also showed activity, and. *H. sericea* follicles are rich in compounds such as lignin (29%), polysaccharides (62.0%), xylose (43.2%) and glucose, (40.5%) and contain a high K concentration [185]. Lipophilic follicle extracts are constituted by bilobol, alkanol and alkanoic saturated acid [185]. The ethanolic aqueous extract of *H. sericea* is composed of total phenolics, flavonoids and condensed tannins [185]. These compounds can be used by hemicellulose fractionation for xylo-oligosaccharide production for medicinal use [185].

Similarly, *H. sericea* biomass can be used for energy production (e.g., biomass combustion and gasification), although its high N and Cl contents do not allow its densification in wood pellets [186]. The C content of *H. sericea* is about 60% and higher and lower heating values of 20.45 and 19.17 MJ.kg^−1^, respectively [186]. Over 75% of *H. prostrata* foliar extract is composed of amino acids such as aspartic acid, glutamic acid, asparagine and glutamine [187]. Additionally, the high amino acid content found in *Hakea* species tissues may indicate that *H. sericea* is a possible source of these compounds for commercial use. Additionally, research on the *H. sericea* tissue mineral content is relevant for assessing potential uses of plants removed in mechanical control operations. The high S content of the plant tissues of *Hakea* species suggests that these minerals could be used for agricultural purposes as a soil biological fertilizer [188]. According to Yusiharni and Gilkes, *H. prostrata* ashes contain high amounts of the water-soluble elements (mg.kg^−1^) Na (8658), Mg (6650), K (9419), Ca (23.43), P (57), B (18) and Cl (9296) and crystalline compounds such as calcite, sylvite, apatite, quartz, medium scolecite, portlandite, lime, and periclase [188]. Most P is in the water-soluble apatite form and is slowly released as an available plant nutrient [188]. Apatite from *H. prostrata* can be applied to acidic soils to increase pH and electrical conductivity, increasing soil fertility. Additionally, *H. prostrata* ash deposition can improve liming acidic soils after prescribed fires, affecting soil properties [188]. Although scarce and directed to other *Hakea* species, research already conducted indicates that there are potential ways of economically valorizing *H. sericea* that should be pursued in the short term.

#### 3.3.7. Synthesis

A synthesis of knowledge on *H. sericea* control methods and the published monitoring and modeling research can be found in Table 7.

## 4. Materials and Methods

### 4.1. Question Formulation

In this review we conducted a systematic review adapted from Pullin and Stewart [189]. It consists of a strict methodological process that guarantees repeatability and reliability [25]. Additionally, we adopted the Preferred Reporting Items in Systematic Review and Meta-Analyses (PRISMA) guidelines to ensure transparency in our research [190]. We started with the formulation of three questions:What ecological and environmental factors (biotic and abiotic, morphological and ecophysiological aspects of the *Hakea* genus and species) contribute to the establishment of *H. sericea* populations?What is the response of *H. sericea* to fires in native and invaded ecosystems?What monitoring and control methods have already been used in *H. sericea* invasion management?

The questions were clearly defined by the subject (*H. sericea*, at different taxonomic/ecological scales: genus and species, individual and population (question 1); population, community and ecosystem (question 2); ecosystem and landscape (question 3)); intervention (scientific support for public policy development and management; monitoring and control actions to reduce *H. sericea* populations in invaded ecosystems); and outcome elements (decrease in *H. sericea* populations in invaded ecosystems).

Specific inclusion criteria were adopted for the questions as follows: for question 1, any ecological factor that influences the adaptive and invasive processes of *Hakea* populations in any geographic area, including their center of origin; for question 2, any ecological, ecophysiological or reproductive response of *Hakea sericea* to fire occurrence in their native habitat or in any geographical area; for question 3, any monitoring, modeling, planning and control interventions (mechanical, chemical, biological or a combination thereof) to mitigate *H. sericea* impacts on invaded ecosystems.

### 4.2. Search Strategy

The search string selection was based on the objective and questions of the review. For the literature search, a holistic approach was adopted, involving a higher number of variables and strings. Firstly, we used the genus in the search strings. After the first selection, publications were selected based on their relevance to the three formulated questions. Filters were applied to remove corrections and additions to previously published papers, and duplicated results were also excluded. After analyzing the title and abstract, publications on analytical chemistry, toxicology, biochemistry and zoology relative to other Proteaceae and *Hakea* species that did not fit the formulated questions were eliminated. All the results for search strings and databases were independently exported to a spreadsheet by two review authors (T.K.B.J and J.C.A), with information on authors, study type (research, review and systematic review, field, laboratory or observational experiment), main questions and outcomes, plant species, ecological scale (individual, population, community, ecosystem, landscape, global), study location (country), date, and DOI (Digital Object Identifier). All publications were associated with one or more of the formulated questions. The search was performed in March and April 2022, using English idiom and the following strings:*Hakea*;*Hakea sericea*;*Hakea sericea* AND fire;*Hakea sericea* AND invasion;*Hakea sericea* AND fire AND invasion;*Hakea sericea* AND fire AND invasion AND management;*Hakea sericea* AND fire AND invasion AND Portugal;*Hakea sericea* AND ecophysiology;*Hakea sericea* AND control.

The databases searched were ISI Web of Science, Scopus, and CAPES Periodicals (Brazil). Capes Periodicals was used to look for publications not in Scopus or ISI Web of science but indexed in other databases (for example, SciELO—Scientific Electronic Library Online).

## 5. Final Remarks

*H. sericea* combines several strategies, such as leaf sclerophylly, low transpiration rates, high stomatal conductance, low hydraulic conductivity potential, seeds with long-distance dispersion and elevated nutrient reserves, and strong serotinous lignified follicles that are resistant to herbivory and pathogens and that are advantageous for the species in invaded ecosystems. *H. sericea* also has high-contact-surface proteoid roots, which allow a high nutrient use efficiency, low tissue construction cost, high P resorption, and high photosynthetic rates that contribute to its adaptation to new environments. Additionally, post-fire response strategies that capitalize on high seed survival rates in heat-resistant follicles improve *H. sericea* competitiveness in fire-prone ecosystems.

All these competitive abilities converge into plant traits underpinning the adaptation and stabilization *H. sericea* in ecosystems where it is introduced, affecting ecosystem structure, composition and function (including fire regime), which impacts ecosystem management, often with negative impacts on the local economy through effects on livestock, grazing, hunting, and tourism and recreation. *H. sericea* control methods, such as felling followed (or not) by prescribed fires, biological control and integrated management, have shown satisfactory results in South Africa, even considering their high monetary cost.

Traditionally, research has focused on the topics covered by the research questions addressed in this review. Novel fields of research that are expected to expand in the following years include the valorization of *H. sericea* products and by-products, such as biomass for energy, molecules with bioactive or phytochemical potential, extracts and ashes from combustion for use as soil fertilizer, as well as the use of the plant in the bioremediation of contaminated soils in areas of invasion. Additionally, research on dynamic monitoring and modeling techniques and tools and the combination of results in integrated approaches to prevent *H. sericea* invasions and to assess control method efficacy is ongoing and expected to expand in the short term due to the demand from stakeholders and decision makers for sustainable and cost-effective invasion control solutions.

## Figures and Tables

**Figure 1 plants-12-00751-f001:**
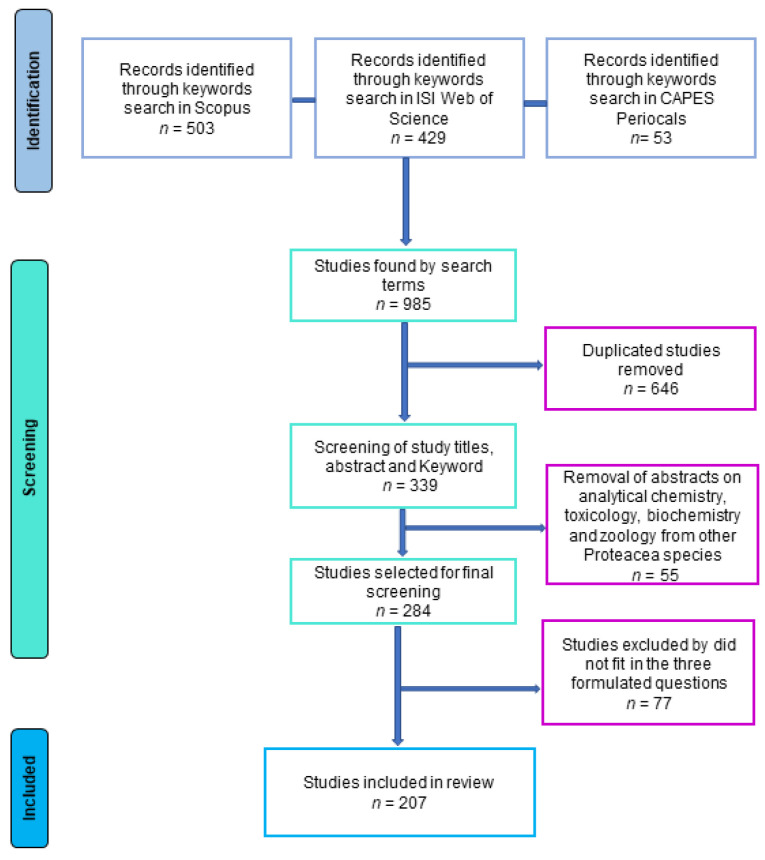
Preferred Reporting Items in Systematic Reviews and Meta-Analyses (PRISMA) flowchart of publications selected in the systematic literature search.

**Figure 2 plants-12-00751-f002:**
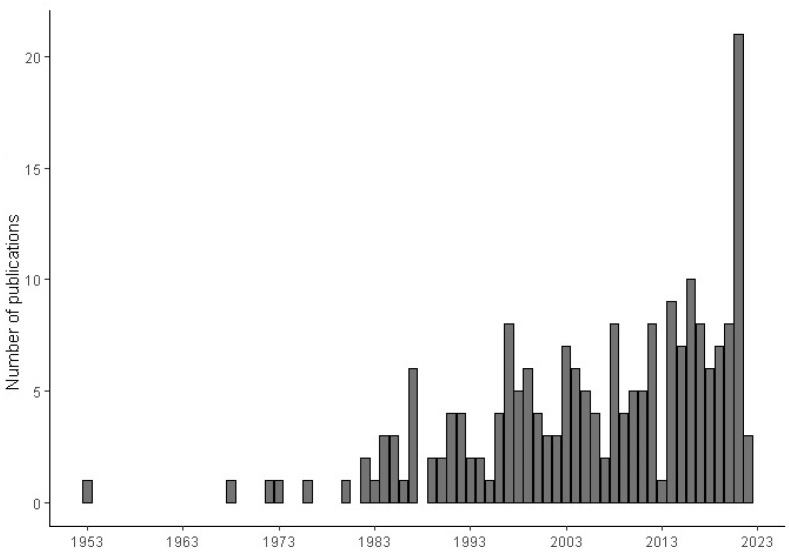
Number of publications per year. Data for 2022 is incomplete since the search was conducted in March and April 2022.

**Table 1 plants-12-00751-t001:** Number of publications (n^o^.) obtained by search string (1 to 9), and total by database.

	Search String									
	1	2	3	4	5	6	7	8	9	TOTAL
SCOPUS	271	62	72	23	11	4	1	3	56	503
Web of Science	239	52	53	18	7	3	1	4	52	429
CAPES periodicals	19	5	11	9	1	1	0	2	5	53
Overall										985

Search strings: 1. *Hakea*; 2. *Hakea sericea*; 3. *Hakea sericea* AND fire; 4. *Hakea sericea* AND invasion; 5. *Hakea sericea* AND fire AND invasion; 6. *Hakea sericea* AND fire AND invasion AND management; 7. *Hakea sericea* AND fire AND invasion AND Portugal; 8. *Hakea sericea* AND ecophysiology; 9. *Hakea sericea* AND control.

**Table 2 plants-12-00751-t002:** Publications (no. and %) related to the formulated questions 1 to 3. See Section 4.1 for details on questions.

Question(s)	Publications	
	No.	%
1	98	47.35
2	24	11.59
3	54	26.09
1, 2	12	5.80
1, 3	13	6.27
1, 2, 3	6	2.90
Total	207	100

**Table 3 plants-12-00751-t003:** Publications (no. and %) including *H. sericea*, *Hakea* sp. and spp., and other *Hakea* species.

	No.	%
Publications including *Hakea sericea*	114	55.1
Publications including unspecified *Hakea* sp. or *Hakea* spp.	23	11.1
Publications including other *Hakea* species	70	33.8
Total	207	100

**Table 4 plants-12-00751-t004:** Publications (no. and percentage (within parentheses)) per question formulated and country. See Section 4.1 for details on questions.

Question(s)	Country					
	South Africa	Australia	Portugal	Global	New Zealand	Other
1	7 (13.2)	67 (66.3)	3 (25)	6 (66.7)	0	15 (57.6)
2	2 (3.7)	16 (15.8)	1 (8.3)	0	4 (66.6)	1 (3.9)
3	39 (73.5)	2 (1.9)	6 (50.0)	0	1 (16.6)	6 (23.1)
1, 2	0	10 (9.9)	0	1 (11.1)	0	1 (3.9)
1, 3	4 (7.5)	4 (3.9)	1 (8.3)	1 (11.1)	1 (16.6)	2 (7.6)
1, 2, 3	1 (1.9)	2 (1.9)	1 (8.3)	1 (11.1)	0	1 (3.9)
Total	53 (25.6)	101 (48.7)	12 (5.8)	9 (4.4)	6 (2.9)	26 (12.6)

**Table 5 plants-12-00751-t005:** A synthesis of the major evolutionary and ecophysiological traits and implications for invasiveness of *H. sericea*.

Group	Trait	Implications
Evolutionary history (*Hakea* genus)	*Hakea* long evolutionary history (35 MY)	High genetic diversity, high phenotypic plasticity, high niche amplitude, widespread biogeographic range
Reproductive strategies	Dense and persistent canopy seed bank	High fecundity ability, high reproductive fitness
Water use efficiency	Leaf sclerophylly	Low transpiration rates
Drought resistance	High stomatal conductance, low hydraulic conductivity potential	Prevention of hydric stress, resistance to aridity, high photosynthetic capacity
Seeds and follicles	Seeds with water retention mechanism, germination in low water potential and in high temperature range	High environmental adaptability and plasticity, resistance to aridity
	High nutrient reserves in seeds cotyledon	Adaptability to dystrophic soils
	Winged seeds, anemochoric dispersion	Long distance dispersion
	High follicle production, thick and lignified follicles	Dense canopy seed bank, resistance to dry conditions, high resilience in response to fire
Serotiny	Strong serotiny, seed retention in follicles up to 10 years	Avoidance of seed predators, adaptation to fire-prone ecosystems, maintenance of dense canopy seed bank
Seedling establishment	High concentration of phenolic compounds	Seedling resistance to herbivory and pathogens
	High nutrient cotyledon content	Fast seedling foliar expansion
Proteoid roots	High contact surface	High P and Mn uptake
	Organic acids/enzymes exudation	Increasing nutrient uptake (inaccessible to other plants)
Nutrient use efficiency	High P use efficiency	Low tissue construction cost, high photosynthetic rates
	High P resorption to new leaves	Adaptation to dystrophic environments

**Table 6 plants-12-00751-t006:** A synthesis of *H. sericea* responses to fire.

Group	Trait	Implications
*H. sericea* post-fire responses and strategies	Large heat-resistant follicles	Seed protection to high temperatures, high seeds survival rates, even after plant death
	High post-fire seed production	High probability of survival, high post-fire recruitment in fire-prone ecosystems
Seed bank/post-fire recruitment	Accumulation of seed storage with increasing fire intervals	High germination rates from seeds in dead plants follicles
	Post-fire seed liberation	Enhanced post-fire population establishment

**Table 7 plants-12-00751-t007:** A synthesis of control methods and knowledge on monitoring and modeling of *H. sericea* invasions.

Group	Type	Implications
Control methods	Mechanical cleaning usually followed by prescribed fires (at different fire intervals)	Widespread and effective depending on access to invaded areas, negative impacts of fire on soil and biodiversity
	Chemical (herbicides)	Expensive and difficult to apply, impact in non-target species; high tolerance due to *H. sericea* thick cuticle, immersed stomata and lower stomatal densities
	Biological (insects)	Coleoptera and lepidoptera species contributed to *H. sericea* population decrease in South Africa
	Biological (Fungi and bacteria)	Development and use of bioherbicides and used as part of sustainable weed management, risk of migrating to other non-target (susceptible) plants
Modeling and monitoring of *H. sericea* invasions	Modeling of invasion factors (biotic and abiotic)	Identification and selection of biotic and abiotic limiting variables, prediction of spread behavior and trends
	Modeling of control agent (biological) impacts on plant fecundity	Prevision of seed load and population growth rates, dispersal distances and new invasive foci
	Modeling costs and benefits	Optimal cost–benefit addressing biodiversity conservation and provision of ecosystem services and invasion reduction in priority areas
	Modeling of *H. sericea* population dynamics in fire risk and control scenarios	System dynamics of control efficacy evaluation in non-planned fire scenarios to be used in decision making process
	Modeling of fire scenarios	Different fuel scenarios and fire behavior and its impacts on ecosystem functioning and services

## Data Availability

Data is contained within the article.

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
