# Peer review of "Invasiveness, Monitoring and Control of Hakea sericea: A Systematic Review"

_plants, 2023, doi:10.3390/plants12040751_

Round 1

Reviewer 1 Report

Dear Authors

Please see attached review comments.

Regards

Reviewer 2 Report

This review aims to summarize, describe and discuss the processes and mechanisms related to Hakea sericea invasion previously reported in the literature, as well as the current knowledge about its monitoring and control. The manuscript is well written and fits well within the scope of the Special Issue “Invasive woody plants – Ecology and Management”. However, my major concern arises from the very low degree of novelty. The traits that explain the superiority of Hakea sericea over native species are well described and reported in the literature, and several reviews are available (please see references 8, 9, 11 and 18). Thus, to be accepted, the authors should highlight and stress what makes this review different from those already available. I think that it would be better if this review contains less information on Hakea species, and discuss in details the data presented in Hakea sericea studies.

Moreover, other points should be clarified:

1.       The terms used to search for publications on several aspects relevant for Hakea sericea should be restricted to this species, since the title and the Abstract only focus this invasive species.

2.       If the general aim of this review is to present the aspects relevant for the success of the Hakea sp., the paper should be reformulated and, in this case, it should include a list of all Hakea species, explaining their differences (for example, type of leaves, reproduction, dispersal, etc.). It seems to be somewhat confused some sections focused on Hakea sp. and other on Hakea sericea. The first question relies most the genus than on Hakea sericea, and the third question is almost entirely dedicated to this species. Why this difference?

3.       In total, the authors obtained 207 papers, but how many papers are exclusively on Hakea sericea? In my opinion, all papers should appear in the text. On contrary, there are a lot of them with general considerations that I am doubt about its importance.

4.       Why the authors used “Capes Periodicals” as a database? Did you find different papers? The same question is applied to “Global Invasive species data-base - https://www.issg.org”.

5.       The search on web databases did not include the terms “monitoring” or “modelling”. Is there any particular reason?

6.       The inclusion of “European ecosystems” as a keyword should be explained.

7.       The impacts of Hakea sericea invasion should be better described.

8. The discussion regarding the future work is unconvincing. This it probably the most innovative part of the review, but only contains general ideas.

PPlease see the attached pdf of the manuscript with other comments and edits.

Round 2

Reviewer 2 Report

Dear Authors,

I agree with your point of view, that this systematic review addresses “simultaneously several aspects of the species”, and, then, this ms constitutes a new approach. As you stated, there are other documents that described/ reviewed each topic and you tried to compile all data in one document. I also agree that some documents are old, but, curiously you used the same old documents in your review (because there is no novel data!). So, your justification for this systematic review does not convince me.

Moreover, I continue with the same doubts concerning the number of documents obtained during the revision process. You obtained 207 papers but not all appear in the text. Why? It seems that not of all are important! In relation to the number of papers focused on Hakea sericea (144), I do not agree with the way to classify these papers. If the species is referred in the middle of the paper as an example, it does not mean that the paper is about such species. Please clarify it and revise it accordingly.

Finally, I am not satisfied with your revision. In the PDF, you can find 39 comments, but you only addressed 14. Why? Some of them are important for clarifying the text, but they were ignored "without explanation". So, at this time, I decided to maintain the first opinion about the ms, i.e., MAJOR REVISIONS.

Round 3

Reviewer 2 Report

Dear Authors,

I carefully read the last version of your manuscript, and I consider that this version was significantly improved. 

I only have a final comment to clarify my opinion that can seem somewhat contradictory: I agree that your manuscript can be viewed as "a new approach", but in some topics I do not consider that your revision complement already existing information, since there are no documents in recent times. 

Apart from this, I believe that your manuscript is now ready to be published in the selected Special Issue.